# High-Resolution PM$_{2.5}$ Concentrations Estimation Based on Stacked Ensemble Learning Model Using Multi-Source Satellite TOA Data

Qiming Fu [1], Hong Guo [2,3,]*, Xingfa Gu [1,2,3], Juan Li [2], Wenhao Zhang [1], Xiaofei Mi [2], Qichao Zhao [1] and Debao Chen [2,3]

1    School of Remote Sensing and Information Engineering, North China Institute of Aerospace Engineering, Langfang 065000, China
2    Aerospace Information Research Institute, Chinese Academy of Sciences, Beijing 100094, China; guxf@aircas.ac.cn (X.G.); lijuan@aircas.ac.cn (J.L.)
3    University of Chinese Academy of Sciences, Beijing 100049, China
*    Correspondence: guohong@radi.ac.cn

**Abstract:** Nepal has experienced severe fine particulate matter (PM$_{2.5}$) pollution in recent years. However, few studies have focused on the distribution of PM$_{2.5}$ and its variations in Nepal. Although many researchers have developed PM$_{2.5}$ estimation models, these models have mainly focused on the kilometer scale, which cannot provide accurate spatial distribution of PM$_{2.5}$ pollution. Based on Gaofen-1/6 and Landsat-8/9 satellite data, we developed a stacked ensemble learning model (named XGBLL) combined with meteorological data, ground PM$_{2.5}$ concentrations, ground elevation, and population data. The model includes two layers: a XGBoost and Light GBM model in the first layer, and a linear regression model in the second layer. The accuracy of XGBLL model is better than that of a single model, and the fusion of multi-source satellite remote sensing data effectively improves the spatial coverage of PM$_{2.5}$ concentrations. Besides, the spatial distribution of the daily mean PM$_{2.5}$ concentrations in the Kathmandu region under different air conditions was analyzed. The validation results showed that the monthly averaged dataset was accurate ($R^2$ = 0.80 and root mean square error = 7.07). In addition, compared to previous satellite PM$_{2.5}$ datasets in Nepal, the dataset produced in this study achieved superior accuracy and spatial resolution.

**Keywords:** satellite remote sensing; PM$_{2.5}$; top of atmosphere; machine learning; ensemble learning

## 1. Introduction

Fine particulate matter with a diameter of less than 2.5 μm (PM$_{2.5}$) can travel long transport distance, having long atmospheric residence times, and that it can pass through the respiratory tract to the depth of the fine bronchial tubes and alveoli. PM$_{2.5}$ is a major source of air pollution and can pose substantial risks to the human. Nepal is a landlocked country in the southern Himalayas, bordered by China and India. The air quality report released by IQAir indicates that Nepal is experiencing severe PM$_{2.5}$ pollution. Nepal was ranked 12th, 10th, and 16th in the world in terms of PM$_{2.5}$ pollution in 2020, 2021, and 2022, respectively. In addition, Kathmandu was 10th, 6th, and 16th in the air pollution rankings of global capitals in 2020, 2021, and 2022, respectively [1–3]. In Nepal, PM$_{2.5}$ pollution is mainly concentrated in the densely populated southern plains region. However, the majority of PM$_{2.5}$ measurement stations in Nepal are concentrated in the Kathmandu area, which is ineffective at detecting continuous spatial and temporal variations in PM$_{2.5}$ concentrations. Therefore, PM$_{2.5}$ concentrations based on satellite remote sensing is worth investigating and has great application potential.

Satellite-based PM$_{2.5}$ concentrations is mainly based on two products: aerosol optical depth (AOD) [4] and top of atmosphere reflectance (TOA). There have been many studies

that have estimated $PM_{2.5}$ concentrations through modeling the $PM_{2.5}$–AOD relationship. However, for Nepal, the currently available AOD products (e.g., the MODIS AOD) have limited spatial coverage and low spatial resolution, and thus cannot meet the demand for fine-grained $PM_{2.5}$ estimation. Different from AOD, TOA products typically have more extensive spatial coverage and higher spatial resolution. Moreover, $PM_{2.5}$ estimation based on TOA products avoids errors in AOD retrieval [5–8].

In recent research, the application of TOA data for $PM_{2.5}$ estimation has become more feasible and practical [5–7]. Many researchers have used various TOA products, such as the MODIS TOA and Himawari-8 TOA, to estimate $PM_{2.5}$ concentrations [8–11]. The existing models for TOA-based $PM_{2.5}$ estimation can generally be categorized into three types: statistical models, machine learning models, and deep learning models. Statistical models typically estimate the $PM_{2.5}$ based on linear relationships between data. Tong et al. [12] utilized the Landsat 8 TOA to establish a combined model that incorporates land use regression and geographically weighted regression. Machine learning models exhibit a stronger nonlinear fitting capability than linear models. Yang et al. [13] applied MODIS TOA data to develop a random forest model for $PM_{2.5}$ estimation in the Yangtze River Delta region of China. Mao et al. [14] established a random forest-based $PM_{2.5}$ estimation model that yielded an $R^2$ close to 0.92. Liu et al. [15] developed an ensemble machine learning algorithm to estimate the $PM_{2.5}$ in China, achieving an $R^2$ value of 0.86. Deep learning models can detect deeper relationships in data, and many scholars have also conducted research on $PM_{2.5}$ estimation and prediction using deep learning models. Yan et al. [16] modeled the Chinese region using the simultaneous ozone and $PM_{2.5}$ inversion deep neural network (SOPiNet), and they verified the performance of the developed model. Yang et al. [17] developed various machine learning and deep learning models to estimate $PM_{2.5}$ concentrations in China. Their results showed that some deep learning models were worse than traditional models. Bai et al. [18] also conducted a comparison of currently popular $PM_{2.5}$ estimation models, and they found that the traditional random forest model outperformed other methods. Ensemble learning models are a type of machine learning model that can leverage the strengths of various models to enhance overall performance. Their effectiveness has been demonstrated in numerous studies [19–22].

With the limited availability of $PM_{2.5}$ measurements, the spatiotemporal distribution of $PM_{2.5}$ concentrations in Nepal remains uncertain. This study introduces a novel stacking model that utilizes Gaofen-1/6 and Landsat-8/9 TOA data, as well as meteorological and auxiliary data. This model was applied to construct a monthly average $PM_{2.5}$ dataset for Nepal.

## 2. Data

### 2.1. $PM_{2.5}$ Measurements

OpenAQ is an air quality data platform dedicated to sharing global air quality data. In this study, the ground-level $PM_{2.5}$ measurements in Nepal were taken from the OpenAQ dataset. The dataset contains two types of data. The first type of data was collected at reference monitoring stations, in which data are usually measured using standardized instruments to ensure accuracy and comparability. While the second type of data is obtained from air sensor stations, which are maintained by individuals or non-government organizations, and use portable or small air sensors to conduct measurements. The hourly measurements were averaged to obtain daily measurements. In total, 8135 samples were obtained from 2018 to 2022. Moreover, Air Pollution in the World (APW) is a platform that provides air quality index (AQI) data around the world. AQI is a standardized index proposed by the United States Environmental Protection Agency (EPA). It can be expressed as:

$$AQI = \frac{I_{high} - I_{low}}{C_{high} - C_{low}}(C - C_{low}) + I_{low} \, , \tag{1}$$

where $C$ represents the PM$_{2.5}$ concentration, $C_{low}$ is the lower limit of PM$_{2.5}$, $C_{high}$ is the upper limit of PM$_{2.5}$, $I_{low}$ is the index limit corresponding to the lower limit, and $I_{high}$ is the index limit corresponding to the upper limit.

As shown in Figure 1, the PM$_{2.5}$ stations in Nepal are mainly concentrated in Kathmandu, while there are fewer PM$_{2.5}$ observations available from PM$_{2.5}$ stations in the south–central part of the country. As a result, the overall PM$_{2.5}$ distribution in Nepal could not be illustrated based on real PM$_{2.5}$ station measurement. The AQI data were therefore introduced as virtual PM$_{2.5}$ data, which could be used to illustrate the PM$_{2.5}$ distribution in Nepal. As shown in Figure 2, a polynomial regression model was used to establish the relationship between AQI and PM$_{2.5}$ using 1840 datapoints matched to the two reference monitoring stations. Here, the AQI-PM$_{2.5}$ fitting accuracy $R^2$ was 0.97.

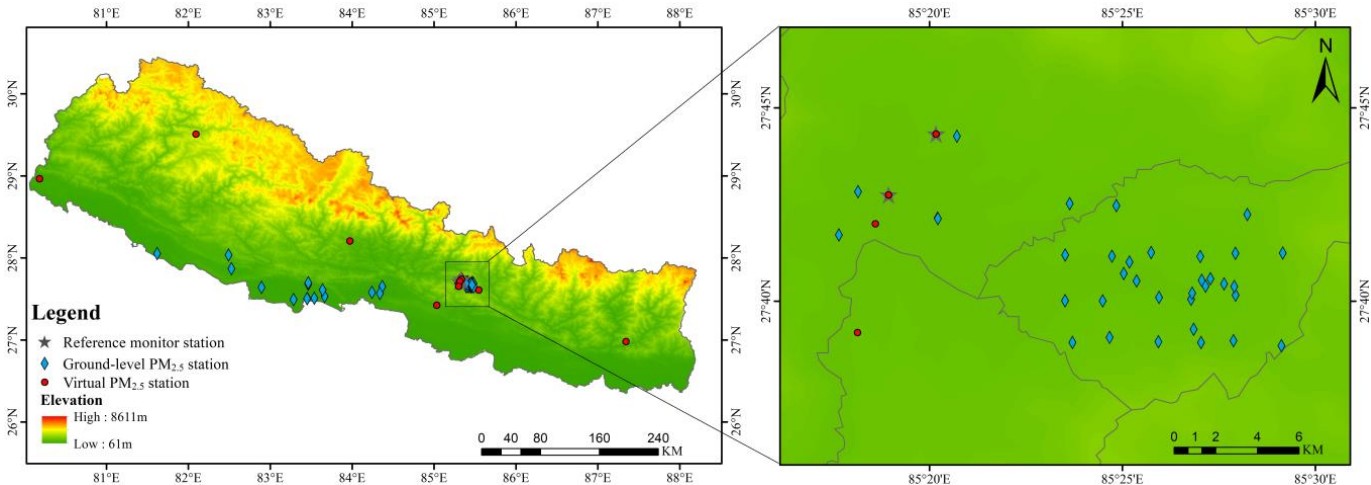

**Figure 1.** Ground-based PM$_{2.5}$ monitoring stations across Nepal.

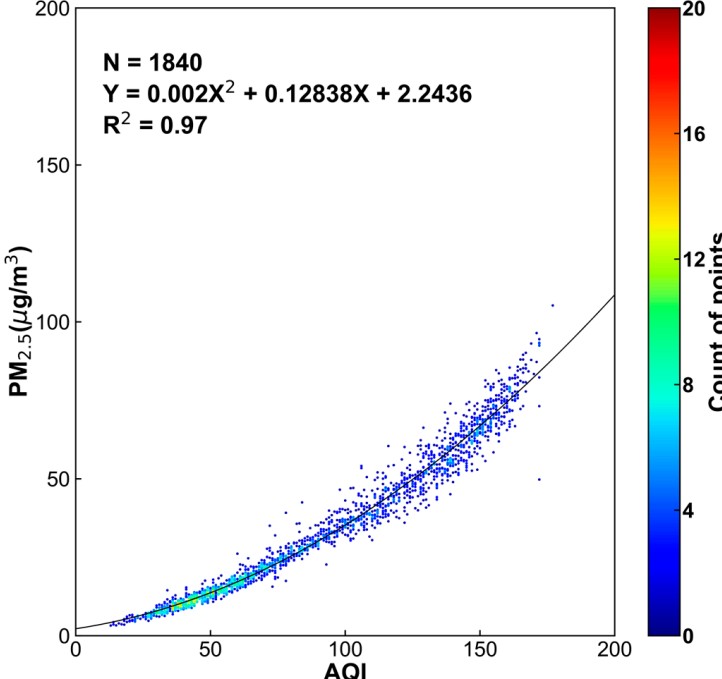

**Figure 2.** Fitment graph of air quality index (AQI) and fine particulate matter (PM$_{2.5}$).

Based on the polynomial regression model, the AQI data from Nepal were corrected to provide "virtual" PM$_{2.5}$ measurements. As shown in Figure 1, the virtual PM$_{2.5}$ mea-

surements were obtained from 10 AQI stations. Virtual daily $PM_{2.5}$ measurements were obtained from 2018 to 2022 for eight of these stations, which had a total of 5037 virtual daily average $PM_{2.5}$ datapoints. Thus, by combining real daily $PM_{2.5}$ measurements (OpenAQ) with virtual daily $PM_{2.5}$ measurements (AQI), this study obtained a total of 13,172 daily average $PM_{2.5}$ values.

### 2.2. TOA Data

In this study, TOA products were obtained from the Gaofen-1/6 satellite and Landsat-8/9 satellite. Gaofen-1/6 data were obtained through the National Remote Sensing Data and Application Service platform. Landsat-8/9 satellite data were acquired through the Google Earth Engine (GEE) platform. The true-color and near-infrared bands of the Landsat-8/9 Operational Land Imager (OLI) data were selected to provide TOA data. Similarly, the true-color and near-infrared bands of the Gaofen-1/6 Wide-Field Camera (WFV) were also selected. However, differences in bands between satellites may introduce uncertainty in the models developed. Detailed information about the spectral bands is shown in Table 1.

**Table 1.** The properties of Landsat-8/9 and Gaofen-1/6 WFV.

| Satellites | Bands | Wavelength (µm) | Spatial Resolution (m) | Temporal Resolution (Day) |
|---|---|---|---|---|
| Landsat-8/9 | Band 2 | 0.45–0.51 | 30 | |
| | Band 3 | 0.53–0.59 | 30 | |
| | Band 4 | 0.64–0.67 | 30 | 16 |
| | Band 5 | 0.85–0.88 | 30 | |
| Gaofen-1/6 WFV | Band 1 | 0.45–0.52 | 16 | |
| | Band 2 | 0.52–0.59 | 16 | |
| | Band 3 | 0.63–0.69 | 16 | 4 |
| | Band 4 | 0.77–0.89 | 16 | |

First, the TOA data from different satellites were resampled to a spatial resolution of 0.001° (100 m) using bilinear interpolation. Second, Landsat 8/9 data with pixel cloud scores above 20 were filtered for quality control. Third, cloud masking of the Gaofen-1/6 data was performed using a thresholding method. Three thresholds were calculated, namely, the RB, the GRB, and the mean feature of the gray-level co-occurrence matrix (GLCM) [23]. The GLCM is a widely used texture feature statistical method introduced by Haralick [24]. The thresholds can be expressed as:

$$RB = \rho_{red} - \rho_{blue}, \tag{2}$$

$$GRB = 4\rho_{green} - \rho_{red} - 3\rho_{blue}, \tag{3}$$

$$Mean = \sum_i \sum_j p(i,j) \times i, \tag{4}$$

where $\rho_{red}$, $\rho_{green}$, and $\rho_{blue}$ represent the radiance values of the red, green, and blue bands, respectively. *Mean* denotes the mean statistical feature of the GLCM, which reflects the regularity of texture in remote sensing images. A $2 \times 2$ sliding window size was employed for the GLCM. Figure 3 illustrates the results of cloud masking for the Gaofen-1 WFV data, indicating that the method can effectively filter out cloudy pixels.

After cloud masking, Landsat-8/9 TOA data can be directly obtained from the corresponding products. For the Gaofen-1/6 WFV images, the digital number (DN) values were first converted into radiance values using Equation (5). Then, the TOA data were obtained from the radiance values using Equation (6). Equations (5) and (6) are calculated as follows:

$$L_\lambda = Gain \times P_{Value} + Offset, \tag{5}$$

$$TOA = \frac{\pi d^2 L_\lambda}{E_0 cos\theta}, \tag{6}$$

where $L_\lambda$ represents the TOA value; $P_{Value}$ is the pixel DN value; *Gain* is the band-specific rescaling multiplier; *Offset* is the band-specific bias; $d$ is the Earth–Sun astronomical unit distance; $E_0$ is the solar irradiance; and $\theta$ is the solar zenith angle.

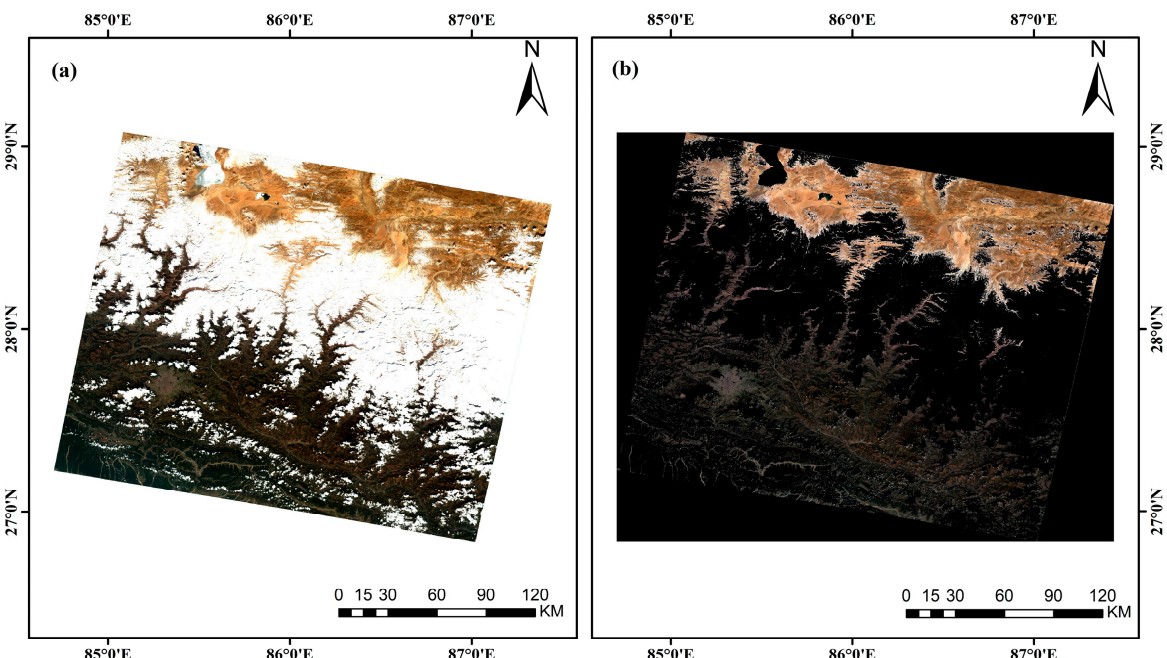

**Figure 3.** Cloud removal effect of Gaofen-1 satellite imagery. (**a**) Image before cloud removal. (**b**) Image after cloud removal.

### 2.3. Auxiliary Data

ERA5 is the fifth-generation atmospheric reanalysis dataset that encompasses uncertainty information for all of the variables at reduced spatial and temporal resolutions [25]. By blending model data with observational data from around the world, ERA5 is a comprehensive and consistent global dataset. In this study, six meteorological data products from ERA5 were included: wind speed (WS), wind direction (WD), 2-m temperature (T2M), relative humidity (RH), boundary layer height (BLH), and surface pressure (SP). Hourly data were obtained for these parameters and the daily average between 8:30 a.m. and 12:30 p.m. was calculated to be consistent with satellite transit times.

The normalized difference vegetation index (NDVI) is a widely used vegetation index. Its values typically range from −1 to 1, and it has the capability to capture background influences on the vegetation canopy, including factors such as the soil type, soil moisture, presence of snow cover, leaf senescence, and surface roughness, as shown in Equation (7):

$$NDVI = (NIR - R)/(NIR + R), \tag{7}$$

where *NIR* represents the near-infrared band value, and *R* represents the red band value. The NDVI data used in this study were calculated based on the TOA data obtained from the Gaofen-1/6 and Landsat-8/9.

The population data (POP) used in this paper were derived from the LandScan dataset. The LandScan dataset provides global population distribution data created by combining geographical spatial science, remote sensing technology, and machine learning algorithms [26].

The Global Multi-resolution Terrain Elevation Data 2010 (GMTED2010) dataset [27] is a product created by the United States Geological Survey (USGS). The GMTED2010 dataset provides global coverage of elevation data across the Earth's surface. These parameters are shown as Table 2.

**Table 2.** Description of meteorological and other data.

| Abbreviations | Data Sources | Spatial Resolution | Temporal Resolution |
| --- | --- | --- | --- |
| BLH | ERA5 hourly data on single levels | 0.25° | 1 h |
| RH | ERA5 hourly data on pressure levels | 0.25° | 1 h |
| T2M | ERA5-Land hourly data | 0.1° | 1 h |
| WS | ERA5-Land hourly data | 0.1° | 1 h |
| WD | ERA5-Land hourly data | 0.1° | 1 h |
| SP | ERA5-Land hourly data | 0.1° | 1 h |
| NDVI | Calculation of TOA data | 0.001° | / |
| POP | LandScan | 1 km | 1 year |
| DEM | GMTED2010 | 0.1° | / |

## 3. Methodology

### 3.1. Machine Learning Model

Extreme Gradient Boosting (XGBoost) is an improved forward additive model based on the boosting strategy [28–32]. This model combines multiple weak learners to train a strong learner. XGBoost introduces regularization terms to control model complexity, and it typically uses the squared error loss function for regression problems. During gradient computation, XGBoost calculates the first and second derivatives of the loss function to the predicted values to understand the trend of errors, allowing for the better adjustment of model parameters. It also uses a greedy algorithm to select the optimal split points, with the aim of minimizing the loss function to the greatest extent.

The light gradient boosting machine (LightGBM) is a cutting-edge gradient boosting framework meticulously designed for distributed learning and highly efficient model training, as documented in various studies [33–37]. To alleviate the shortcomings of XGBoost, LightGBM uses the technique of Gradient-based One-Side Sampling. This approach significantly diminishes both the complexities of time and space, while concurrently mitigating the risk of overfitting. Additionally, LightGBM incorporates a leaf-wise growth strategy with a depth limit.

### 3.2. Bayesian Optimization Algorithm

The Bayesian optimization method uses Gaussian processes to continuously update iterations based on parameter information from previous training results [38–42]. This leads to the optimal combination of hyperparameters. The algorithm is based on the historical evaluation results of the objective function, $f(x)$. It establishes a prior distribution and combines the observed points obtained in previous iterations to determine a posterior distribution. This iterative process continually optimizes and ultimately minimizes the objective function, $f(x)$. Bayesian optimization initially assigns values to the model's hyperparameters, where $X = x_1, x_2, \ldots, x_n$, represents the value of a certain hyperparameter. It then uses a sampling function $f(x)$ to determine the next sampling point, as shown in Equation (8):

$$x_t = argmin f(x), x \in X. \tag{8}$$

The hyperparameters of the XGBoost and LightGBM models were optimized using a Bayesian algorithm. Specific parameters for each model are presented in Table 3.

**Table 3.** The parameters for each model.

| Models | Parameters | Values |
|---|---|---|
| XGBoost | n_estimators | 673 |
| | max_depth | 9 |
| | min_child_weigth | 3 |
| | gamma | 0.3 |
| | subsample | 0.87 |
| | colsample_bytree | 0.81 |
| | learning_rate | 0.01 |
| LightGBM | n_estimators | 840 |
| | max_depth | 4 |
| | min_child_samples | 20 |
| | min_child_weight | 0.001 |
| | num_leaves | 31 |
| | colsample_bytree | 1 |

Cross-validation (CV) is a widely used method for assessing the generalization ability and accuracy of models. Therefore, this study employed 10-fold CV to evaluate the model's performance. Ten-fold CV divided the training dataset into 10 parts, with one part used as a validation set during each iteration, and the remaining data used as the training set to train the model. The results from 10 iterations were averaged to obtain a final result. This study evaluated the model using four metrics: the coefficient of determination R-squared ($R^2$), the slope, the root mean squared error (*RMSE*), and the mean absolute error (*MAE*). The specific formulas for these metrics are as follows:

$$R^2 = 1 - \frac{\sum_{i=1}^{n}(y_i - \hat{y}_i)^2}{\sum_{i=1}^{n}(y_i - \overline{y})^2}, \tag{9}$$

$$MAE = \frac{1}{n}\sum_{i=1}^{n}|y_i - \hat{y}_i|, \tag{10}$$

$$RMSE = \sqrt{\frac{1}{n}\sum_{i=1}^{n}(y_i - \hat{y}_i)^2}, \tag{11}$$

where $y_i$ represents the observed value, $\hat{y}_i$ represents the predicted value, and $\overline{y}$ represents the mean of the observed values.

*3.3. Ensemble Stacking Model*

Stacking models usually comprise multiple base learners to develop a meta learner with enhanced stability and generalization [43–45]. These models are also known as heterogeneous ensemble methods. First, $m$ base learners are trained on the original data, resulting in feature data of dimension ($m$, $p$). These data are then fed into the second-level model to obtain the final prediction. For the base learners, models with different structures are often selected to enhance generalization. In this study, an ensemble stacking model was proposed with XGBoost and LightGBM as the first-level models, and linear regression as the second-level model, named XGBLL model. The Bayesian optimization algorithm was employed to fine-tune each model. The flowchart for constructing the XGBLL model is illustrated in Figure 4.

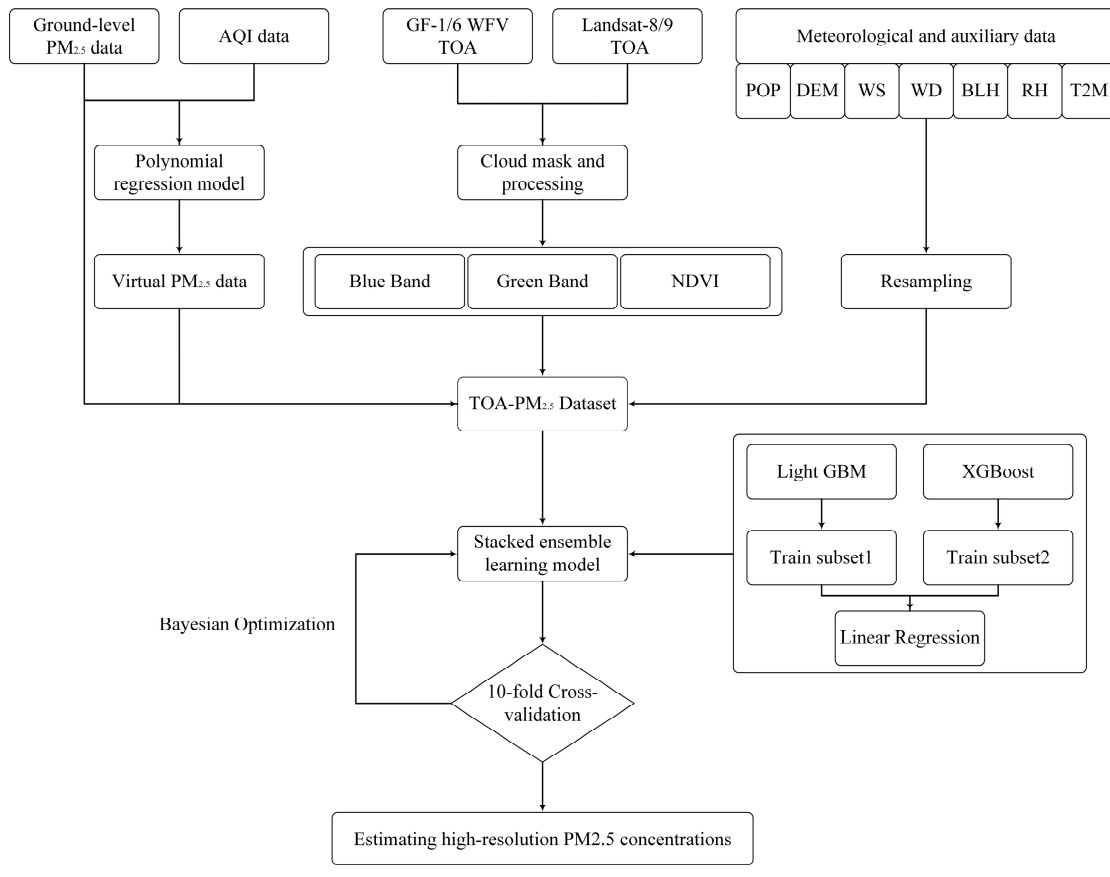

**Figure 4.** Flowchart of fine particulate matter (PM$_{2.5}$) estimation.

## 4. Results and Analysis

### 4.1. Evaluation of the XGBLL model

This study compared single machine learning models with XGBLL model (Figure 5). The results of comparison indicated that the R$^2$ values for the XGBoost and LightGBM models were 0.79 and 0.80, respectively; the RMSE values were 10.74 and 10.39, respectively; the MAE values were 7.48 and 7.16, respectively; and the slopes were 0.75 and 0.80, respectively. The best performance was achieved with the XGBLL model constructed in the present study, which had an R$^2$ of 0.81, an RMSE of 10.28, and an MAE of 7.08. However, the PM$_{2.5}$ values from virtual stations may introduce uncertainties to the developed model.

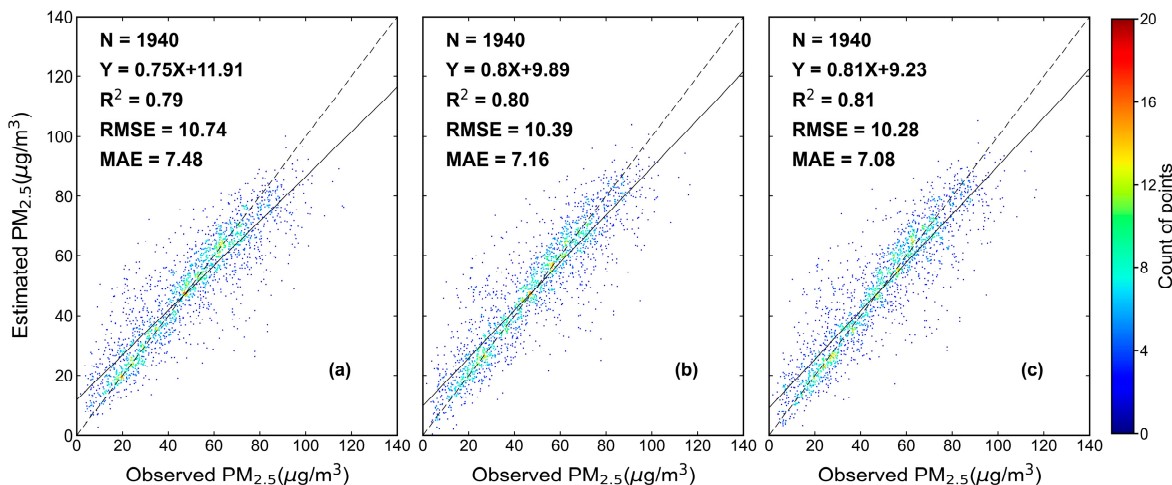

**Figure 5.** Cross-validation results. (**a**) XGBoost model. (**b**) LightGBM model. (**c**) XGBLL model.

### 4.2. High-Resolution PM$_{2.5}$ Concentration Monitoring Application

Kathmandu is surrounded by the Langtang range of the Himalayas and the Himalayas themselves. Kathmandu is about 1400 m above sea level, making it one of the higher cities in Nepal. The city is situated in the Kathmandu Valley, and the terrain is relatively flat. Kathmandu has a temperate monsoon climate with four distinct seasons.

Using the XGBLL model, and based on the TOA data obtained from Gaofen-6, the daily average PM$_{2.5}$ concentrations for the Kathmandu region were derived. The different scenarios for the Kathmandu region are shown in Figure 6. As Figure 6 shows, there is a good air quality in Kathmandu on 2 November 2022, with the PM$_{2.5}$ concentrations was low in most parts of the city (Figure 6a1,a2). However, there is a severe pollution on 21 December 2022, with the PM$_{2.5}$ concentrations were high in the city center, ranging from 90–100 μg/m$^3$ (Figure 6b1,b2). The results showed that the high spatial resolution PM$_{2.5}$ concentration accurately reflected the distribution of PM$_{2.5}$ values in Kathmandu.

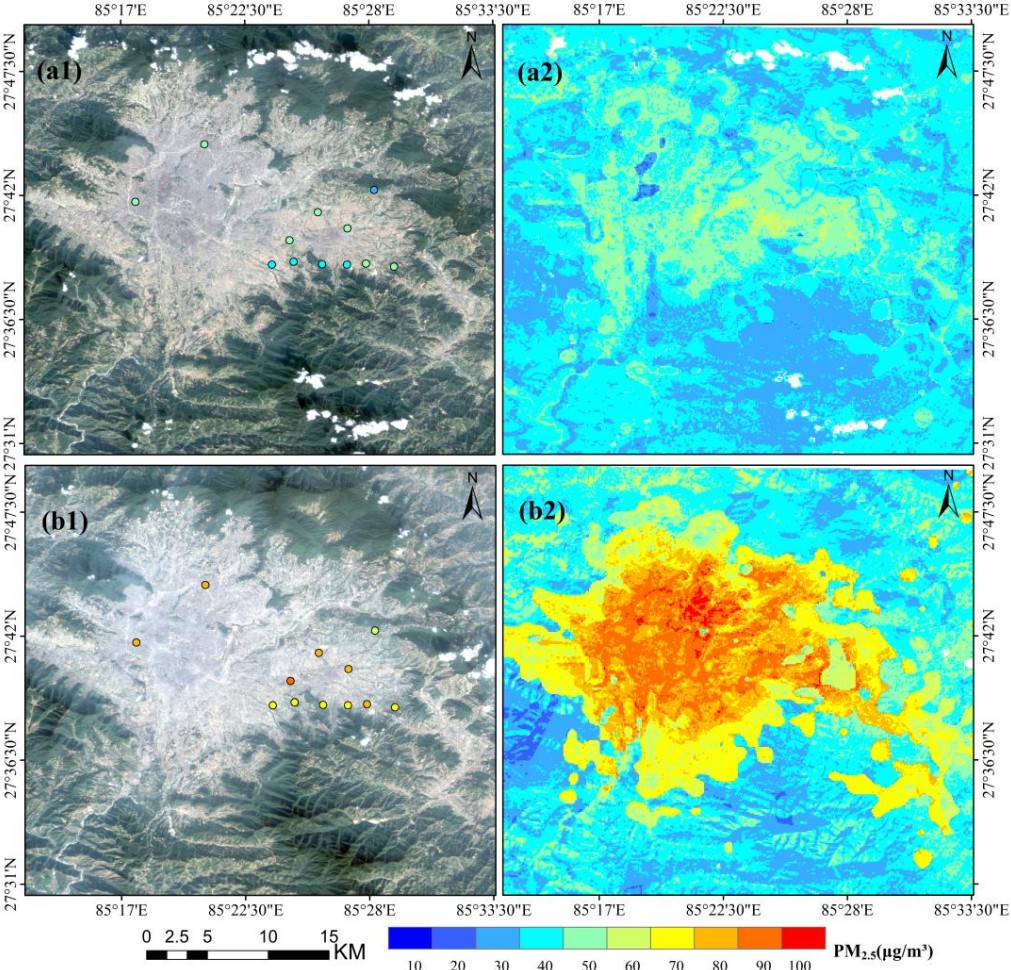

**Figure 6.** Different scenarios for Kathmandu. (**a1**) Gaofen-6 true-color image of Kathmandu area on 2 November 2022. (**a2**) Daily average fine particulate matter (PM$_{2.5}$) in the Kathmandu region on 2 November 2022 as estimated by Gaofen-6. (**b1**) Gaofen-6 true-color image of Kathmandu area on 21 December 2022. (**b2**) Daily average PM$_{2.5}$ in the Kathmandu region on 21 December 2022 as estimated by Gaofen-6.

### 4.3. Fusion of the Nepal PM$_{2.5}$ Dataset

The PM$_{2.5}$ predictions from different satellites were averaged to generate the final prediction. Figure 7 shows the PM$_{2.5}$ estimation results from various satellites in Nepal for February 2020, as well as the final fused result, we can find almost full coverage of PM$_{2.5}$ values in February 2020 (Figure 7d).

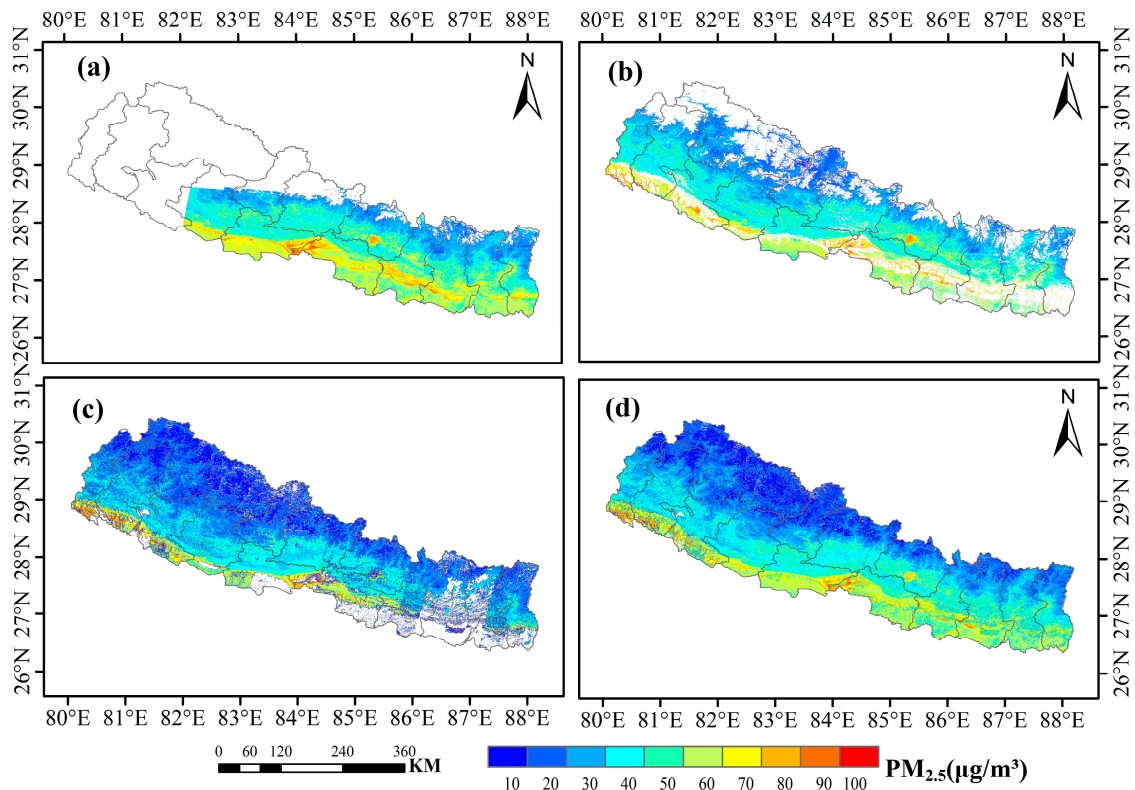

**Figure 7.** Comparison of monthly average fine particulate matter (PM$_{2.5}$) Estimation from various satellite types in February 2020. (**a**) PM$_{2.5}$ map from Gaofen-6. (**b**) PM$_{2.5}$ map from Gaofen-1. (**c**) PM$_{2.5}$ map from Landsat-8. (**d**) Fused PM$_{2.5}$ map.

### 4.4. Nepal PM$_{2.5}$ Dataset Evaluation

To ensure accuracy, the dataset was validated using only data collected from real PM$_{2.5}$ measurement stations and monthly average data selected from ground-level PM$_{2.5}$ stations with at least 20 days of valid data. A total of 15 monthly average PM$_{2.5}$ points for 2020 were selected to validate the dataset. The XGBLL model constructed in this study had an R$^2$ of 0.80, an RMSE of 12.56, and an MAE of 8.69.

Van Donkelaar created the V5GL03 global PM$_{2.5}$ dataset [46]. The dataset combines AOD from the MODIS, MISR, and SeaWIFS satellites with the GEOS-Chem chemical transport model. It then undergoes calibration using geographically weighted regression to improve the accuracy of its estimates. As shown in Table 4, the accuracy of the V5GL03 dataset in Nepal is as follows: R$^2$ = 0.75, RMSE = 18.00, and MAE = 14.06. In contrast, the dataset produced in this paper has the following accuracy for Nepal: R$^2$ = 0.80, RMSE = 12.56, and MAE = 8.69. The dataset generated in this paper exhibits a significant improvement in accuracy for Nepal compared to the V5GL03 dataset.

**Table 4.** The validation results of different PM$_{2.5}$ dataset.

| Dataset | R$^2$ | RMSE | MAE |
| --- | --- | --- | --- |
| V5GL03 | 0.75 | 18.00 | 14.06 |
| This study | 0.80 | 12.56 | 8.69 |

Figure 8 shows the monthly average PM$_{2.5}$ concentrations from V5GL03 and the datasets in the present study. The PM$_{2.5}$ predictions from the XGBLL model offer higher resolution and clearer textural features, enabling a more detailed representation of PM$_{2.5}$ distributions.

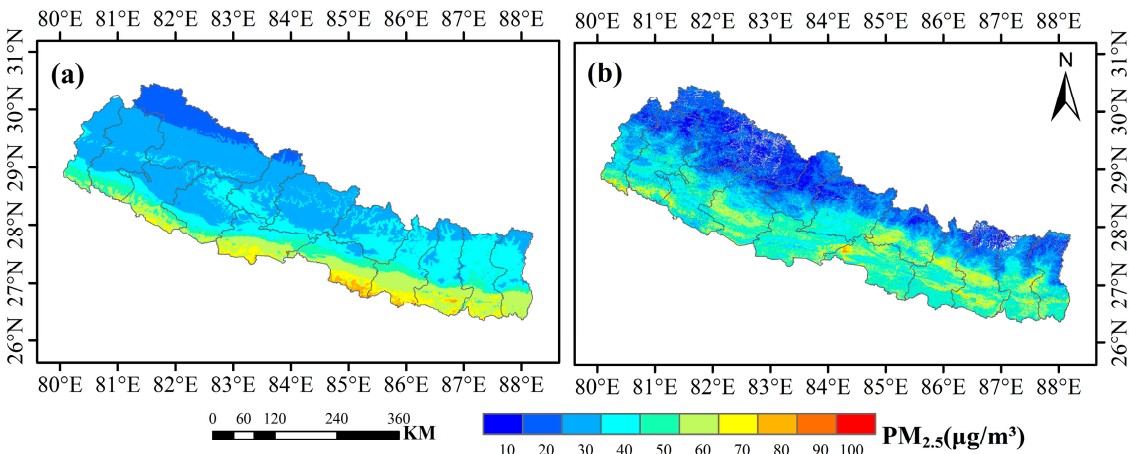

**Figure 8.** Comparison with other datasets. (**a**) Fine particulate matter (PM$_{2.5}$) map for March 2020 (V5GL03). (**b**) PM$_{2.5}$ map for March 2020 created in the present study.

### 4.5. Spatiotemporal Distribution of PM$_{2.5}$ Values in Nepal

Based on the monthly PM$_{2.5}$ dataset developed for Nepal in 2020, this study analyzed the spatiotemporal variation of PM$_{2.5}$ in Nepal at different temporal scales.

Figure 9 illustrates the temporal variations of PM$_{2.5}$ concentrations on a monthly scale. Notably, during 2020, PM$_{2.5}$ pollution in Nepal exhibited distinct fluctuations. Specifically, from January to April, the PM$_{2.5}$ values ranged between 25 and 35 μg/m$^3$. After April, a discernible decline in PM$_{2.5}$ values occurred and continued until July. Then, the PM$_{2.5}$ values began to increase gradually.

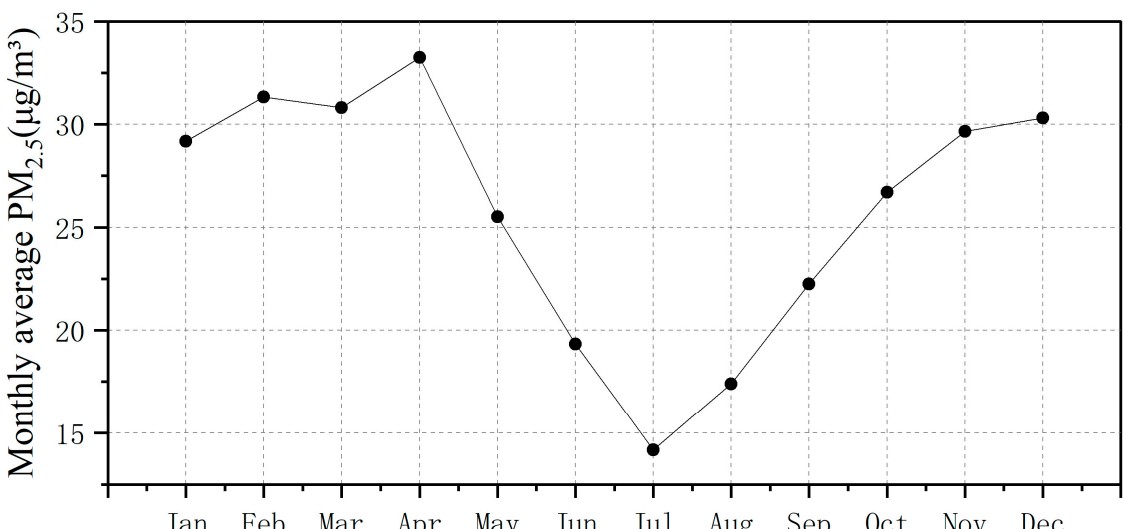

**Figure 9.** Monthly average fine particulate matter (PM$_{2.5}$) variation in Nepal in 2020.

Figure 10 illustrates the spatial distributions of seasonal average PM$_{2.5}$ concentrations across Nepal in 2020. It is worth noting that the northern regions of Nepal, characterized by high altitudes and a sparse population, experienced high air quality, resulting in minimal variation in PM$_{2.5}$ concentrations throughout the year. In contrast, the central and southern regions of Nepal, characterized by hosting the majority of the country's population, exhibited a distinct two-season pattern. During the spring and winter, which is the dry season in Nepal, PM$_{2.5}$ concentrations were high. Subsequently, with the onset of the rainy season, which spans the summer and autumn months, PM$_{2.5}$ pollution significantly decreased.

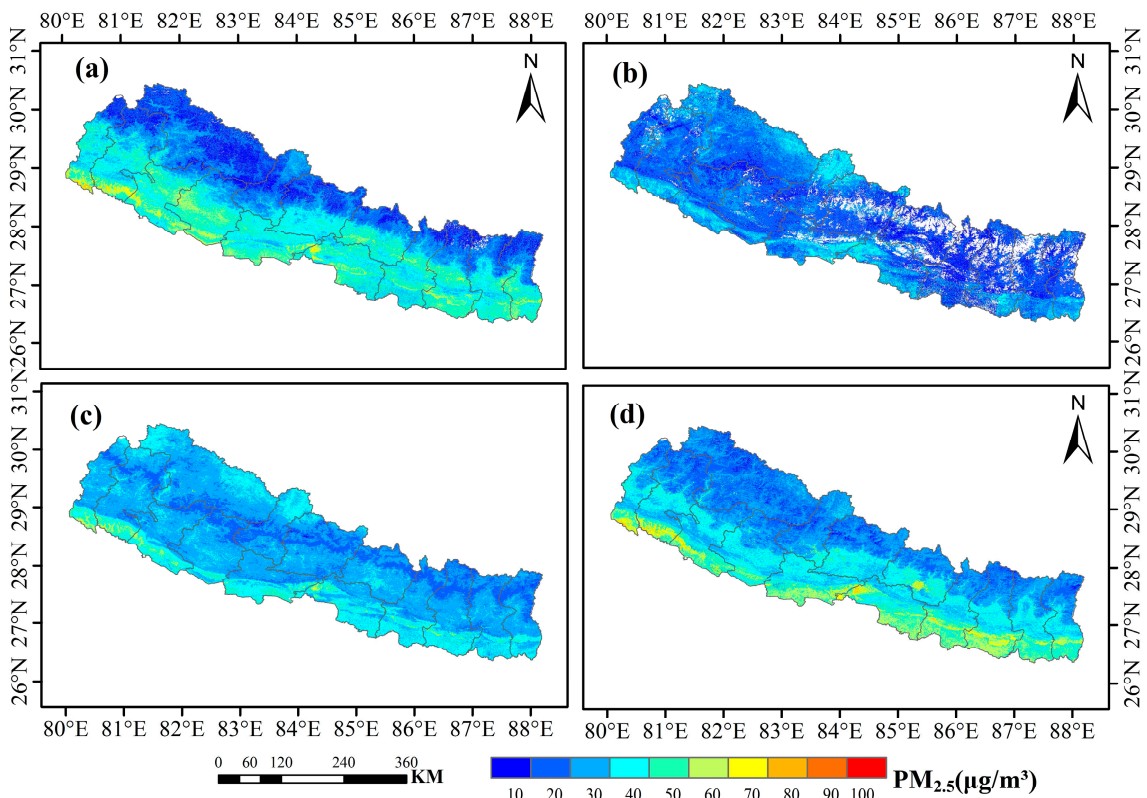

**Figure 10.** Seasonal average fine particulate matter (PM$_{2.5}$) Map in Nepal for the year 2020. (**a**) Spring. (**b**) Summer. (**c**) Autumn. (**d**) Winter.

The spatial distribution of the annual average PM$_{2.5}$ concentration in Nepal is presented in Figure 11. It can be observed that the northern areas exhibit lower PM$_{2.5}$ concentrations, and PM$_{2.5}$ values increased from north to south. The capital city of Kathmandu experienced higher PM$_{2.5}$ values.

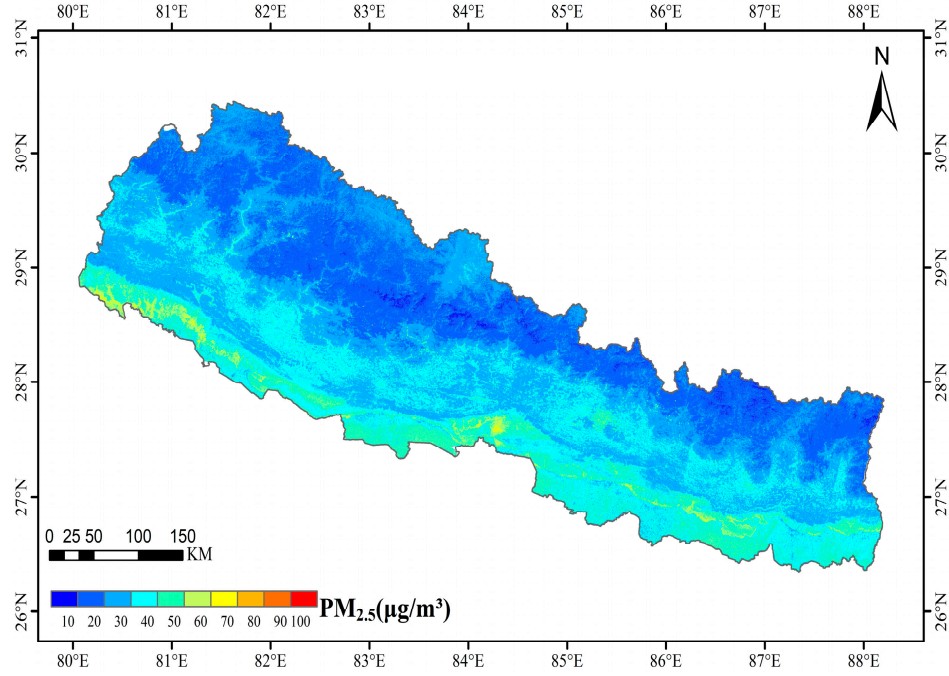

**Figure 11.** Annual average fine particulate matter(PM$_{2.5}$) Map in Nepal for the year 2020.



### 5. Conclusions

This study developed a XGBLL model to estimate the $PM_{2.5}$ concentration in Nepal. The training dataset was extended by introducing AQI data. Various models were fine-tuned using Bayesian optimization to improve performance. We produced daily averaged Nepal $PM_{2.5}$ concentration data from Gaofen-1, Gaofen-6 and Landsat-8, and analyzed the distribution of $PM_{2.5}$ concentration in Kathmandu under different pollution scenarios. In addition, the integration of Gaofen-1/6 WFV and Landsat-8/9 OLI TOA data greatly extended the spatial coverage of $PM_{2.5}$ predictions. The results showed that the XGBLL model achieved higher model accuracies, with an $R^2$ of 0.80, an RMSE of 12.56, and an MAE of 8.69. These results outperform the individual models and provide valuable insights for further research in the field of $PM_{2.5}$ estimation using TOA data, as well as $PM_{2.5}$ estimation using Gaofen data.

**Author Contributions:** Q.F., H.G. and X.G. conceived and designed the experiments; Q.F. and D.C. performed the experiments; J.L. and W.Z. contributed in data processing and data analyses; X.M. and Q.Z. contributed to interpretation of results and critical discussion of findings. All authors have read and agreed to the published version of the manuscript.

**Funding:** This work was supported by the National Key Research & Development Program of China (Grant Number: 2019YFE0126700, 2020YFE0200700), the Natural Science Foundation of China (Grant Number: 42271358), The Major Project of High Resolution Earth Observation System (Grant Number: 30-Y60B01-9003-22/23), and the Common Application Support Platform for National Civil Space Infrastructure Land Observation Satellites (Grant No. 2017-000052-73-01-001735).

**Data Availability Statement:** Data are contained within the article.

**Acknowledgments:** The high-resolution satellite remote sensing data used in this paper were obtained from the National Remote Sensing Data and Application Service platform and the Google Earth Engine platform. Meteorological data products were provided by the European Centre for Medium-Range Weather Forecasts. Population data were sourced from the Oak Ridge National Laboratory of the U.S. Department of Energy. Elevation data was provided by the USGS. Ground-level $PM_{2.5}$ measurement data were obtained from OpenAQ (https://openaq.org/, accessed on 1 November 2023), and ground-level Air Quality Index (AQI) data were sourced from APITW (http://aqicn.org/city/all/, accessed on 1 November 2023). I would like to express my sincere gratitude to these organizations for providing access to the data and resources essential for this research.

**Conflicts of Interest:** The authors declare no conflict of interest.

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
