# Peer review of "High-Resolution PM2.5 Concentrations Estimation Based on Stacked Ensemble Learning Model Using Multi-Source Satellite TOA Data"

_remotesensing, doi:10.3390/rs15235489_

Round 1
Reviewer 1 Report
Comments and Suggestions for Authors
This paper developed a stacked ensemble learning model attempt to estimate high-resolution PM2.5 concentration in Nepal from multi-source satellites with high resolution TOA observations. The result show that the proposed model works better and the spatial distribution of PM2.5 is reasonable. The manuscript is well organized, I recommend the paper can be published after some minor revision.
[1] Like “xxx et al[X].” Remove the dot ”.”;
[2] Line60: two dot behind ”0.92”;
[3] Line90: why not include PM station data around Nepal? For example, Indian.
[4] Line154: where is the NDVI data from?
[5] Fig4: Bayesian Optimization Algorithm is not included;
[6] Line174: “Methodoogy “ should be “Methodology“
[7] Table 3: use a line to separate the XGBboost and ET in the table.
[8] Figure 5: should labeled with a, b, c, and stand for which machine learning model.
[9] In all, make a Moderate revision of the English expression.
Comments on the Quality of English LanguageModerate editing of English language required.
Reviewer 2 Report
Comments and Suggestions for Authors
This paper discusses the estimation method of particulate matter pollution in Nepal from a single ground-based PM2.5 concentration observation station. The spatial and temporal distribution of PM2.5 in Nepal was obtained by using the stacked ensemble learning model, multi-source satellite data, single ground station data and regional AQI data. It is of great significance to improve the awareness of PM2.5 concentration in Nepal. However, there are some problems in this paper, which need to be revised and improved.
1、From the perspective of modeling and analysis of the spatial and temporal distribution of PM2.5, the data from 2020 to 2022 seem to be less. It is recommended to use 5 years of data for analysis, that is, to use the data from 2018 to 2022, so that the model has better applicability.
2、The training samples in this paper are too few and not representative.
3、It should be an attempted method to establish the relationship between TOA radiation and AQI, and then obtain the relationship between TOA radiation and PM2.5 according to the relationship between AQI and PM2.5 concentration. However, the relationship between TOA radiation and AQI obtained in China is related to a variety of factors in the region, as mentioned in the relevant part of the paper, and needs to be supplemented according to the first comment.
4、And extrapolating this relationship to the entire Nepalese region, the corresponding relationship in some regions of Nepal should be inadequate. It is suggested that certain regions of Nepal corresponding to the situation in China should be selected to carry out the extrapolation study of the spatial and temporal distribution of PM2.5 concentration, which is somewhat reasonable.
5、The related bands of these two satellites are inconsistent, how to deal with them?
6、Although the spatial resolution of the satellite is merged to 100 meters, from the application point of view, the impact of the surface with a resolution of 100 meters on TOA radiation is still very large. It is suggested that the resolution should be further sampled to about 500 meters to reduce the uncertainty of the impact of surface reflected radiation on TOA radiation.
7、Line 60,One more point.
8、Line 119,more than 20, Is it written wrong?
9、Line 143,are offset and gain parameters, It's upside down.
10、Table 2, Is the data source of RH written incorrectly?
11、Line 174,Methodoogy, It's misspelled.
12、 Line 243,Figure 5 requires the headings a-c.
13、The English expression level of the paper needs to be further improved, and it is suggested to ask a professional company to polish it.
Comments on the Quality of English LanguageThe English expression level of the paper needs to be further improved, and it is suggested to ask a professional company to polish it.
Reviewer 3 Report
Comments and Suggestions for Authors
The study uses a stacked ensemble learning model to estimate PM2.5 concentrations at 100 m spatial resolution over Nepal, where in situ measurements are sparse. Top of atmosphere (TOA) data comes from GF-1/6 WFV and from Landsat-8/9, which have higher spatial resolution than the TOA aerosol optical depth (AOD) from sensors such as MODIS; the ERA5 reanalysis provides six meteorological parameters and NDVI. The resulting PM2.5 estimate showed an R2 value of 0.74 vs. PM2.5 measured by the single available ground site in Nepal. Accurate PM2.5 estimates on a resolution appropriate to urban scales can be highly important to public health. Although I have concerns about the accuracy of the measurements used for validation, the manuscript is logically organized and well written. My specific comments are below:
Lines 79-84. These two links appear to be mostly low-cost sensor networks, which are useful in aggregate despite the limited capability of individual sites. More detail about these data sources would be helpful. But since only one OpenAQ site is used in this study, is there more information available about that specific sensor?
Lines 94-96. The composition of aerosols likely differs between China and Nepal. Does this affect the AQI-PM2.5 relationship?
Figure 1. Is there a way to indicate the density of scatterplot points on this plot (e.g. with a color scale)?
Lines 245-247. Have you considered whether different satellites should be weighted differently in the averaged result? Do individual satellites have different validation statistics?
Round 2
Reviewer 2 Report
Comments and Suggestions for Authors
The authors have revised and improved this paper in accordance with the review comments. This paper is recommended for publication.